# Effect of Exergame Training on Working Memory and Executive Function in Older Adults

**Chenxi Zhao** [ID]**, Chenglei Zhao** [ID]**, Minmin Zhao, Lin Wang, Jiawei Guo, Longhai Zhang, Yunfeng Li** [ID]**, Yuliang Sun** [ID]**, Ling Zhang, Zheng'ao Li and Wenfei Zhu** *[ID]

School of Physical Education, Shaanxi Normal University, Xi'an 710119, China
* Correspondence: wzhu@snnu.edu.cn; Tel.: +86-177-9232-1530

**Abstract:** Background: As the population ages, cognitive impairment and dementia have become one of the greatest health threats in older adults. Prior studies suggest that exergaming could improve cognitive function in older adults. To date, few long-term exergames intervention studies on older adults during the COVID-19 epidemic exist. This study aimed to investigate the effects of exergame on cognitive function in Chinese older adults, and to examine whether exergame was more effective than aerobic dancing for executive function and working memory. Methods: 55 participants (mean age = 65.4 ± 3.7 years) were randomly assigned to an exergame training (ET) group, an aerobic dancing training (ADT) group, or a control (CON) group. The ET and ADT groups received 36 sessions (three 75-min training sessions per week, exercise intensity = 65 to 75% $HR_{max}$) during a 12-week period. The outcome measures for cognitive function included working memory measured by the N-back test, and executive function measured by the Stroop test. Results: The ET group showed a significantly positive effect in working memory, relative to the ADT (accuracy in 1-back test: ES = 0.76, $p < 0.01$), and CON group (accuracy in 1-back test: ES = 0.87, $p = 0.02$). Moreover, the performance in the Stroop test showed some improvements in executive function after intervention in the ET and ADT groups (Stroop intervention effect: ES = 0.38; $p = 0.25$). Conclusions: Exergame had a positive benefit in improving cognitive functions in older adults without cognitive impairment. Long-term exergame training could improve working memory in older adults. Exergame and aerobic dancing can efficiently improve inhibitory control of executive function in older adults. Maintaining an active lifestyle is protective of cognitive health in older adults.

**Keywords:** exergame; working memory; executive function; older adults

## 1. Introduction

Currently, physical activity was reduced amongst older people because of the COVID-19 home confinement, while physical fitness and cognitive function declined with aging [1,2]. The latest statistics in China have shown that approximately 15 million people over the age of 60 have dementia and 9.8 million have Alzheimer's disease (AD), and the prevalence of mild cognitive impairment (MCI) in people over age 60 is as high as 15.5% [3]. However, there is still no treatment or cure for dementia, and it is crucial to prevent the risk of this disease in older adults without apparent cognitive impairment.

The evidence suggests that potentially modifiable lifestyle factors may influence cognitive health in later life and offer the potential to reduce the risk of cognitive decline and dementia [4,5]. Increasing physical activity is of great significance in improving the cognitive function of older adults [6–8]. Previous research has shown that a variety of types of exercise training can improve the emotional and cognitive function of older adults [9–11]. Even an acute bout of moderate-intensity aerobic exercise can significantly improve inhibitory control behavioral performance in older adults [12]. Long-term aerobic exercise has shown effectiveness in delaying the development of MCI [13]. Similarly, long-term resistance exercise was effective in improving the processing speed and executive function in older adults [14].

Exercise combined with cognitive training can be considered as cognitive-motor dual-task training. Cognitive-motor dual-task training has been shown to improve gait, balance, and cognitive function in older adults and in stroke or Parkinson's patients [15–18]. The combining of cognitive and physical exercise training has been demonstrated to be effective in improving the cognitive function in healthy older adults or those with MCI or dementia [19,20]. It can add cognitive benefits to executive function and working memory, compared to sequential cognitive and physical exercise training and exercise alone [21,22].

Exergaming can be used to deliver cognitive-motor dual-task training. Exergame is a product that has become popular in the past decade combining cognitive tasks in gameplay with significant physical exercise by using physical input devices [23]. Previous meta-analyses suggested that exergame was confirmed to improve overall cognition in older adults [24,25]. The social elements and interesting content of the exergame can enhance the older adults to have a continuous and positive motivation for exercise training, which would obtain additional exercise benefits [26]. Most of the recent studies verified the effects and feasibility of exergame training in older adults by comparing them with the control group. However, under a period of normalization of prevention and control for the COVID-19 epidemic, no similar long-term intervention studies in older adults have been found. Additionally, fewer studies have compared the effects of exergames and other forms of exercise on cognitive function.

Although the benefits of exercise training on cognitive function in older adults are well confirmed, the type of exercise for obtaining optimal cognitive benefits of specific cognitive domains remains unknown. The effect of exergame training on working memory and executive function in older adults is still unclear. The purpose of this study was to investigate the effects of exergame on cognitive function in Chinese older adults. We also examined whether exergame was more effective than aerobic dancing in executive function and working memory. We hypothesized exergame had positive effects on executive function and working memory in this population, and could even be more beneficial than traditional aerobic dancing.

## 2. Materials and Methods

### 2.1. Study Design and Participants

Participants aged 60 or older were recruited from Shaanxi Province in China. The mini-mental state examination (MMSE) was used to identify participants with cognitive impairments. The physical activity readiness questionnaire (PAR-Q) was used to identify participants with potential health risks associated with exercise. Individuals were included if they: (a) did not have cognitive impairment; (b) did not have dyskinesia due to orthopedic or neurological problems; (c) did not have cardiopulmonary disease; and (d) did not have a severe visual impairment. All the participants were informed of the purpose of our study, and written informed consent was obtained. The study was approved by the scientific Ethics Committee at the Shaanxi Normal University (202116003). Baseline information was collected from the baseline survey and physical examination, including age, sex, race, region of residence (whether living in urban), education, smoking, alcohol drinking, and health status.

Of the 82 volunteers who were recruited, 69 participants were eligible to participate in the study. They were randomly assigned to one of three groups, namely an exergame training group (ET, n = 25), an aerobic dancing training group (ADT, n = 22), and a no-training control group (CON, n = 22). The ET and ADT groups received exercise intervention supervised by our team every week for 12 weeks. The CON group was asked not to change their exercise habits and maintain regular activities.

Of the 69 participants enrolled in the study (Figure 1), 14 participants did not complete the study: 7 (1 in ET, 6 in CON) withdrew due to health reasons, and 7 (3 in ET, 4 in ADT) withdrew due to scheduling conflicts. Finally, 55 participants (ET n = 21, ADT n = 18, and CON n = 16) completed all exercise interventions and cognitive assessments. Thus, data for 55 participants were used in the final statistical analysis.

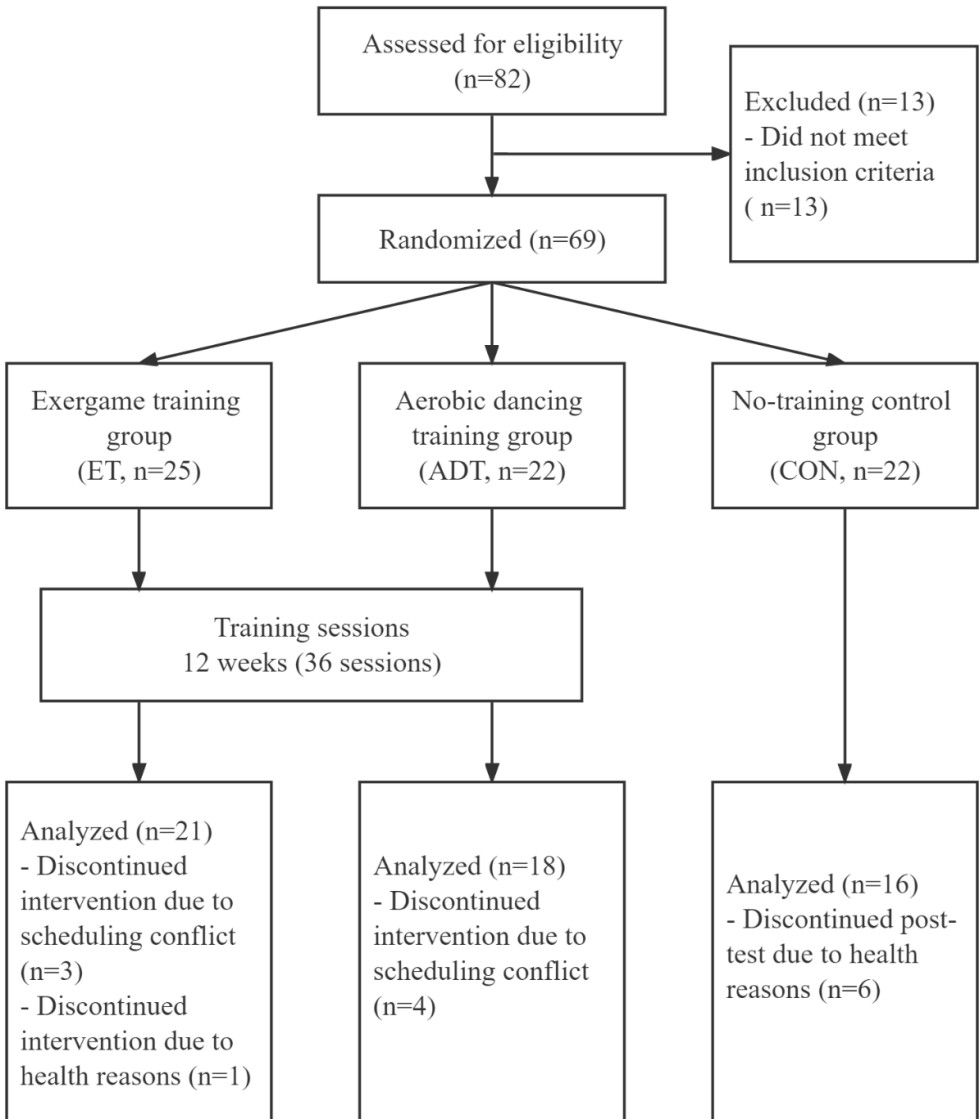

**Figure 1.** Overview of the study design.

### 2.2. Intervention

Participants in two intervention groups completed a 12-week training program with three sessions per week. Each session lasted for 75 min (a 10 min warm-up, 60 min formal training, and 5 min relaxation exercises). Participants wore heart rate monitors (Fit-mao, Shandong Delay Cook Health Care Equipment Co., Dezhou, China) during the training to ensure that the exercise intensity ranged from 65 to 75% of the participant's maximal heart rates ($HR_{max} = 206.9 - 0.67 \times$ age) [27].

The training for the ET group was conducted in an indoor laboratory. Before the exergame training course, we conducted a pre-experiment to estimate the intensity of the exergame for the older adults. According to the pre-experiment results, the moderate intensity exergames were suitable for older adults. Training programs consisted of Fitness Boxing 2, Zumba, and Mario Tennis Ace (Figure 2) which used the Nintendo Switch system (Nintendo Co., Kyoto, Japan). Fitness Boxing 2 is a virtual boxing game. Participants held the joy-con to control their players using different boxing movements. Zumba is a dance game that consists of different music styles. Participants followed a virtual dance instructor to dance and received feedback through the joy-con. Each dance was selected according to its intensity and acceptability to the participants. Mario Tennis Ace is a virtual tennis game. Participants held the joy-con as a racket and controlled the player to hit the ball

under the swing mode. Four to six participants were included at the same time with two Switch systems working together. The trainer demonstrated the operation of the switch device and three exergames and guided them to understand how to use the switch device and play the exergame. Participants adjusted their movements according to instantaneous visual and joy-con feedback.

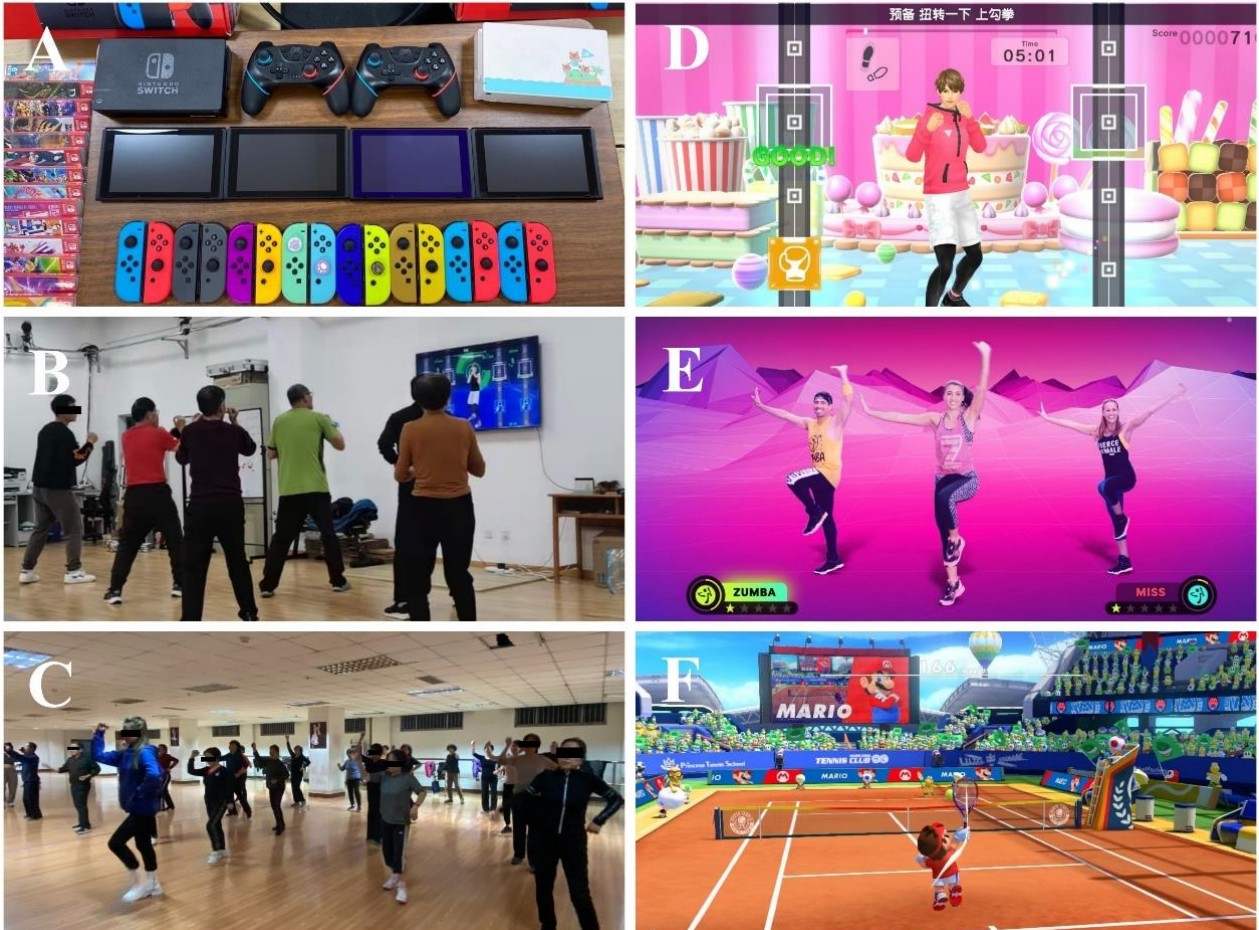

**Figure 2.** The exergaming setup and training program: (**A**) Nintendo Switch system; (**B**) Exergame training (ET) participants; (**C**) Aerobic Dancing Training (ADT) participants; (**D**) Fitness Boxing 2; (**E**) Zumba; (**F**) Mario Tennis Ace.

The training for the ADT group was conducted in an indoor dancing room. Three sets of soul-sexy jazz dance routines were designed according to the acceptable movements of the older adults and preset exercise intensity. Twenty-two participants were included at the same time in dance courses on Monday, Wednesday, and Friday. The ADT group's exercise intensity was moderate, the same as that of the ET group.

Each training session was supervised by 1–2 physical trainers. We motivated each participant to make an effort when they did not reach 65% to 75% $HR_{max}$ according to their heart rate. If a participant reached an exercise intensity above the pre-determined range, they were asked to take a short rest.

### 2.3. Measures

Two cognitive tasks were designed to test participants' working memory and executive function, respectively. All stimulation presentation and data from the cognition tests were performed by e-prime 3.0 software (Psychology Software Tools, Sharpsburg, PA, USA).

### 2.3.1. N-Back Test

Participants were tested by two versions of N-back tasks: the 0-back test as the control condition and the 1-back test as the low working memory load condition [28]. The stimulus of the test was a number randomly selected from a set of numbers 1–9 displayed in the center of the screen. In the 0-back condition, the number "7" was defined as the target. In the 1-back condition, the target was any number identical to the number presented in the 1 trial before. Each block with 24 trials (8 targets) was repeated once. Each block lasted 60 s and was interspersed with 20 s resting blocks [29]. In each trial, the stimulus presentation time was 500 ms. The blank interval was set to 2000 ms. Two short tasks of two conditions were practiced before the formal test, and the participants can have a 30 s rest before starting the official test. The design of the N-back test is presented in Figure 3.

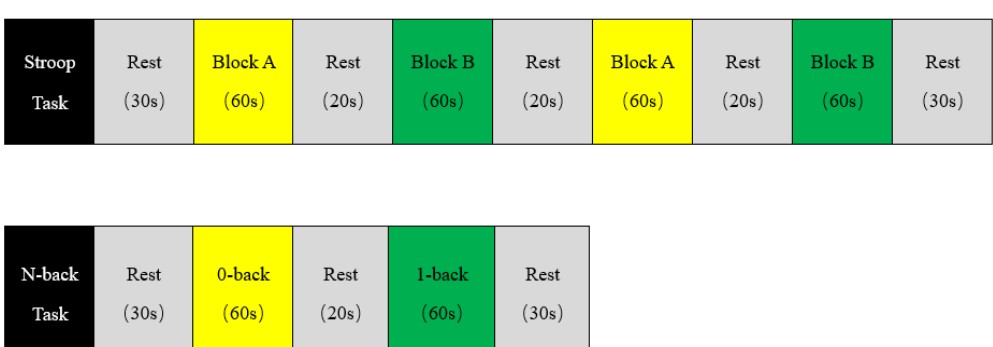

**Figure 3.** Experiment paradigm in block design for cognitive tasks (Stroop task, and N-back task).

### 2.3.2. Stroop Test

A modified Stroop color task [30] (Figure 3) was used to measure executive functions. Participants were required to identify the color of the stimulus in each of two conditions, and then provided their responses with two fingers on a ZM keyboard. Block A was the color-congruent condition, and the stimulus was a color name written in the same color ink, such as the Chinese word meaning "red" printed in red ink. Block B was the incongruent condition, and the stimulus was a color name printed in different ink colors, such as the word "green" printed in red ink. All conditions included two colors (red and green). Each block with 15 trials was repeated to appear twice. Each block lasted 60 s, and was then interspersed with 20 s resting blocks. In one trial, before the stimulus arose, a fixation point "+" displayed for 0.5 s, and the visual stimuli appeared in the center of the computer screen for 1.5 s. The blank interval appeared for 2 s. The test consisted of a shorter practice with two conditions before the formal test to make participants understand the task.

### 2.4. Statistical Analysis

The behavioral data collected by the e-prime software was imported into Excel for preprocessing. Individuals who missed selections were filled with his/her personal average value. If the correct rate of reaction time was <100 ms, data were marked as errors and filled with the personal average value [31]. Personal reaction time values exceeding mean ± 3 SD were replaced by the individual mean values of the same condition.

Data were analyzed with Statistical Package for Social Sciences software (SPSS Version 25.0, IBM Corporation, Armonk, NY, America). Behavioral performance on the N-back and Stroop test was indicated by the accuracy (ACC), reaction time (RT), and Stroop interference effect ($SIE = RT_{blockB} - RT_{blockA}$) [32]. Normality and homogeneity of variance were tested ($p > 0.05$). In order to find out the effect of the 12-week exergame on cognitive function among older adults, the changes in cognitive function before and after 12 weeks (post values minus pre values) were tested between ET and ADT, as well as ET and CON by the independent-samples t-test, respectively. If data were not normally distributed, Mann–Whitney U tests were used instead. Additionally, Cohen's effect sizes (ES) were calculated to compare the magnitude of differences in cognitive function between the

groups. Cohen's thresholds of 0.2, 0.5, and 0.8 were interpreted as small, moderate, and large effects, respectively [33]. The level of significance was set to $\alpha = 0.05$. Previous studies [34,35] have reported that a *p*-value depends on the sample size. Different sample sizes may lead to different conclusions. However, the effect size is not affected by sample size and can express the expected differences in the outcome between two interventions or conditions.

## 3. Results

Table 1 presents the demographic characteristics of participants at baseline. The average age of participants was $65.4 \pm 3.7$ years. Among the 55 participants, 72.7% were women, and 92.7% were living in urban. The ADT group had more women than other groups. Most participants were high school graduates (32.7%). The rates of smoking were low in all groups. The rate of alcohol drinking in the ET group was higher than in other groups. The score in MMSE was higher in the ET group than in other groups.

**Table 1.** Participant demographics.

| Variable | ET Group (N = 21) | ADT Group (N = 18) | CON Group (N = 16) | Total (N = 55) |
|---|---|---|---|---|
| Age, mean (SD), year | 65.2 (3.7) | 65.2 (4.3) | 65.8 (3.2) | 65.4 (3.7) |
| Female, n (%) | 14 (67.7) | 16 (88.9) | 10 (62.5) | 40 (72.7) |
| Region of residence, n (%) | | | | |
| Urban | 20 (95.2) | 18 (100.0) | 13 (81.3) | 51 (92.7) |
| Rural | 1 (4.8) | 0 (0) | 3 (18.7) | 4 (7.3) |
| Education, n (%) | | | | |
| Less than high school | 8 (38.1) | 3 (16.7) | 3 (18.8) | 14 (25.5) |
| High school graduate | 5 (23.8) | 6 (33.3) | 7 (43.8) | 18 (32.7) |
| College for professional training graduate | 6 (28.6) | 5 (27.8) | 2 (12.5) | 13 (23.6) |
| College graduate and above | 2 (9.5) | 4 (22.2) | 4 (25.0) | 10 (18.2) |
| Smoking, n (%) | 1 (4.8) | 0 (0) | 1 (6.3) | 2 (3.6) |
| Alcohol drinking, n (%) | 3 (14.3) | 1 (5.6) | 1 (6.3) | 5 (9.1) |
| MMSE score, mean (SD) | 29.0 (1.05) | 28.7 (1.08) | 28.7 (1.14) | 28.8 (1.08) |

ET, Exergame Training Group; ADT, Aerobic Dancing Training Group; CON, No-training Control group; MMSE, The Mini-Mental State Examination.

Small to moderate effects were observed for the ET group in both N-back conditions, relative to the ADT group (ACC 0-back: ES = 0.22, $p = 0.86$; ACC 1-back: ES = 0.76, $p < 0.01$; RT 0-back: ES = 0.24, $p = 0.46$; RT 1-back: ES = $-0.51$, $p = 0.12$). Moderate to large effects were observed for the ET group in 1-back, relative to the CON groups (ACC 1-back: ES = 0.87, $p = 0.02$; RT 1-back: ES = $-0.52$, $p = 0.13$) (Table 2).

**Table 2.** The effects of the 12-week intervention on working memory measured by the N-back tests.

| | | ET GROUP (N = 21) | | ADT GROUP (N = 18) | | CON GROUP (N = 16) | | ET vs. ADT | | | | ET vs. CON | | | |
|---|---|---|---|---|---|---|---|---|---|---|---|---|---|---|---|
| | | 0 month | 3 month | 0 month | 3 month | 0 month | 3 month | t/U | p | d | ES | t/U | p | d | ES |
| ACC [a] | 0-back | 93.7 (9.3) | 97.6 (3.6) | 95.1 (6.4) | 97.2 (4.7) | 94.8 (5.6) | 97.9 (4.6) | 183.0 | 0.86 | 0.22 | S | 155.0 | 0.68 | 0.10 | - |
| | 1-back | 79.5 (14.2) | 88.4 (10.2) | 89.4 (9.9) | 88.6 (8.6) | 87.2 (13.5) | 86.1 (14.0) | 97.5 | <0.01 ** | 0.76 | M | 386.0 | 0.02 * | 0.87 | L |
| RT [b] | 0-back | 544.6 (97.8) | 543.9 (118.8) | 540.6 (86.6) | 516.1 (58.8) | 532.3 (89.1) | 521.3 (76.2) | 0.74 | 0.46 | 0.24 | S | 0.34 | 0.74 | 0.11 | - |
| | 1-back | 713.3 (182.7) | 651.7 (126.3) | 662.2 (119.6) | 666.9 (150.7) | 663.4 (171.0) | 660.5 (151.2) | −1.58 | 0.12 | −0.51 | M | −1.6 | 0.13 | −0.52 | M |

ET, Exergame Training Group; ADT, Aerobic Dancing Training Group; CON, No-training Control group; ACC, accuracy; RT, reaction time. ES, effect size: d = 0.2–0.5, small effect; d = 0.5–0.8, medium effect; d > 0.8: large effect. [a] Mann–Whitney U tests; [b] independent-samples t-test. * $p < 0.05$ indicates a significant difference between ET and ADT or ET and CON. ** $p < 0.01$ indicates a significant difference between ET and ADT or ET and CON.

A small effect existed in ACC for the ET group for the color-congruent condition, relative to the ADT group (ES = 0.43; *p* = 0.20), whereas a medium effect existed for the ET group when compared to the CON group (ES = 0.54; *p* = 0.16). A small effect existed in RT for the color-congruent condition and in SIE for the ET group when compared to the ADT group (ES = −0.26, 0.38, respectively; *p* = 0.43, 0.25, respectively) (Table 3).

**Table 3.** The effects of the 12-week intervention on executive function measured by the Stroop tests.

| | | ET GROUP (N = 21) | | ADT GROUP (N = 18) | | CON GROUP (N = 16) | | ET vs. ADT | | | | ET vs. CON | | | |
|---|---|---|---|---|---|---|---|---|---|---|---|---|---|---|---|
| | | 0 month | 3 month | 0 month | 3 month | 0 month | 3 month | t/U | p | d | ES | t/U | p | d | ES |
| ACC [a] | color-congruent condition | 98.1 (3.9) | 100.0 (0) | 98.9 (3.2) | 98.3 (6.3) | 98.8 (2.1) | 98.8(2.4) | 157.0 | 0.20 | 0.43 | S | 131.0 | 0.16 | 0.54 | M |
| | color-incongruent condition | 95.6 (11.4) | 98.4 (2.3) | 96.5 (8.7) | 98.5 (2.3) | 92.9 (24.9) | 97.7 (5.0) | 177.0 | 0.70 | 0.08 | - | 140.0 | 0.31 | −0.12 | - |
| RT [b] | color-congruent condition | 557.4 (121.6) | 536.7 (84.7) | 533.1 (68.0) | 538.0 (95.0) | 566.5 (89.4) | 543.4 (110.8) | −0.80 | 0.43 | −0.26 | S | 0.06 | 0.95 | 0.02 | - |
| | color-incongruent condition | 654.1 (179.4) | 622.6 (141.3) | 628.2 (127.2) | 593.4 (111.3) | 633.9 (140.6) | 615.1 (171.4) | 0.08 | 0.94 | 0.03 | - | −0.25 | 0.80 | −0.08 | - |
| SIE [b] | | 96.7 (90.0) | 85.9 (79.6) | 95.1 (92.7) | 55.4 (68.9) | 67.4 (86.5) | 71.7 (98.4) | 1.18 | 0.25 | 0.38 | S | −0.46 | 0.65 | −0.15 | - |

ET, Exergame Training Group; ADT, Aerobic Dancing Training Group; CON, No-training Control group; ACC, accuracy; RT, reaction time; SIE, Stroop intervention effect. ES, Effect-size: d = 0.2–0.5, small effect; d = 0.5–0.8, medium effect; d > 0.8: large effect. [a] Mann-Whitney U tests; [b] independent-samples *t*-test.

## 4. Discussion

The purpose of the present study was to determine whether the 12-week exergame training has cognitive benefits for older adults. The main findings of the study were: (a) 12-week exergame can significantly improve working memory in older adults; (b) exergame was more effective on working memory than aerobic dancing training in this study setting. Findings from this study can establish a better system of exergame training for improving older adults' cognitive function to help attenuate the compounding effects of the COVID-19 pandemic on the physical inactivity and age-related cognitive decline in older adults.

In our study, the exergame was effective in improving working memory compared with the control group. An interesting finding was that the exergame training might be a better alternative to improve working memory, because exergame was more effective on memory than aerobic dancing training. Compared with the baseline, the accuracy of the post-test 1-back task in the exergame group was significantly improved, and the reaction time was significantly decreased, whereas the results in the control group and the aerobic dancing group were reversed. Emerging evidence suggests that exergame may improve working memory in older adults. A randomized-controlled study has investigated the change in cognitive functions in healthy older adults using exergame and balance training, and participants who completed 24 exergame training showed a significant decrease in the reaction time during the working memory task [36]. Another study showed a 12-week exergame training might have the potential to maintain and improve cognitive function, particularly in short-term and long-term memory in older adults [37]. In frail older adults, Liao et al. [38] found that only the exergame group showed significant improvements in verbal and working memory after a 12-week intervention. A previous report which measured working memory by N-back task indicated that a large improvement was found in working memory after 4-week motor-cognitive dual-task training, and that working memory scores also improved significantly (large ES = 1.38) [15]. There is no similar study on the different effects of exergame and aerobic exercise on working memory. To our knowledge, this is one of the first randomized controlled trials to investigate the differential effect of exergaming versus aerobic dancing training on cognitive function in older people.

The beneficial effect on working memory after exergame training may be explained by recent knowledge. On the one hand, working memory load is associated with executive

attention and distraction avoidance [39]. Participants need more attention maintaining under 1-back task. Previous studies have reported exergame significantly improves attention span and attentional processing in older adults [25,40]. The exergaming activities used in our study included fitness boxing and tennis playing, which need the simultaneous coordination of movements and cognitive activities, and the recall of movements and attention processing. Therefore, participants in the exergame group had better cognitive performance under more working memory load conditions. On the other hand, the dorsolateral prefrontal cortex (DLPFC) structure and function play a central role in working memory [41,42], and sustained attention [39]. A functional near-infrared spectroscopy recordings (fNIRS) study showed that acute cycling exercise enhanced the activation of the left dorsolateral prefrontal cortex (DLPFC) and left orbital frontal cortex (OFC). Therefore, increased oxygen availability for cerebral metabolic activity is a possible neurophysiological mechanism underlying the benefits of exercise on working memory. However, there is no direct evidence that exergame has a positive effect on DLPFC function. Future work can combine fNIRS to explore the effects of exergame training on brain function.

Compared with the control group, exergame is more beneficial to inhibitory function in older adults. There is no significant difference in the effect of exergame on executive function compared with aerobic dancing exercise, and the effect may be slightly weaker than that of aerobic exercise. This finding is consistent with those of recent studies. A meta-analysis had reported that exergame had a significant influence on overall executive function in older adults, and exergaming was more effective on the executive function when the intervention duration was long [24]. Similarly, a study [43] showed that 12-week exergaming-based tai chi can improve the performance of the Stroop Color and Word Test in MCI patients, and exergame was comparable to traditional tai chi for the enhancement of executive function. On the contrary, some studies have not found an improvement in the cognitive function of older adults after exergame intervention [44]. In a recent study [45], researchers from the Netherlands found 12-week exergame and aerobic exercise intervention had no effects on executive function, episodic memory and working memory in older adults with dementia. Different results in the effect of exergame training may be due to the dose–response relationship between exercise and cognition. Further work is needed to clarify the dose–response effects of exergame on cognitive function and their relationship with healthy older people.

Improvement in executive function, especially in those who get exergame and aerobic dancing training, may result from the greater functional activation in the left inferior frontal gyrus after exercise. Executive function is associated with brain activation in the left inferior frontal [46]. A previous study investigated the regional brain changes in functional activation by a whole-brain fMRI scan and reported greater left inferior frontal activations after exercise training [47]. In our study, greater left inferior frontal activations may enhance functional processing while engaging in task conditions requiring greater amounts of executive control. Furthermore, a positive mood during exercise may influence the benefit of exercise on prefrontal activation and executive performance [48]. Research has shown exercise with music leads to greater enhancement of a positive mood. Fitness Boxing 2 and Zumba with music used in exergame intervention was a continuous combination of music and exercise to better stimulate positive emotions in the participants.

This study had several strengths. First, a randomized controlled trial was used to explore the effects of different types of exercise on cognitive function in older adults. We also set a control group to explore the actual effects of exercise intervention. Second, this is one of the first studies to apply the Nintendo Switch device on the effect in the exergame intervention study. In previous studies, Nintendo Wii and Microsoft Xbox 360 Kinect were used to intervene in exergame games. The Nintendo Switch system is an upgraded version of the Nintendo Wii system, and it has more games and is more convenient. Our research expands the research model of exergame intervention. Third, the cognitive function was objectively measured by e-prime 3.0 software.

Several limitations of our study should also be considered. First, the cognitive tasks adopted block design, and participants may predict answers according to previous trials in the same block while doing the Stroop task. Second, the sample size was relatively small, and the control group was not an attention control group. The time spent engaged could have a differential impact on participants' cognitive function between groups. Third, a longer period of intervention or follow-up may be necessary to better understand the effects of exergame training on cognitive function. Finally, the selection of cognitive assessments is limited. Further work should explore the impact of exergame on more cognitive domains.

## 5. Conclusions

Our study found that exergame had a positive effect on cognitive function in older adults without cognitive impairment. Long-term exergame training can improve working memory in older adults. Exergame was more effective on working memory than aerobic dancing training. Exergame and aerobic dancing both can efficiently improve the inhibitory control of executive function in older adults. Maintaining an active lifestyle is protective of cognitive health in older adults. Findings in our study can establish a better system of exergame training for improving older adults' cognitive function to help attenuate the effects of the COVID-19 pandemic on the age-related cognitive decline in older adults.

**Author Contributions:** Conceptualization, C.Z. (Chenxi Zhao) and W.Z.; methodology, Y.L., W.Z. and C.Z. (Chenxi Zhao); formal analysis, W.Z., C.Z. (Chenxi Zhao), M.Z., C.Z. (Chenglei Zhao) and L.Z. (Longhai Zhang); investigation, Y.S. and C.Z. (Chenxi Zhao); writing—original draft preparation, C.Z. (Chenxi Zhao), W.Z. and C.Z. (Chenglei Zhao); writing—review and editing, M.Z., L.W., J.G., L.Z. (Ling Zhang), Y.S., Z.L. and W.Z.; supervision, W.Z.; funding acquisition, W.Z. All authors have read and agreed to the published version of the manuscript.

**Funding:** This study was supported by the MOE (Ministry of Education in China) Project of Humanities and Social Sciences (20YJC890053), and Shaanxi Province Social Science Foundation Program (2020Q009).

**Institutional Review Board Statement:** The study was conducted according to the guidelines of the Declaration of Helsinki, and approved by the Academic Committee of Shaanxi Normal University (202116003; 15 July 2021).

**Informed Consent Statement:** Informed consent was obtained from all subjects involved in the study.

**Data Availability Statement:** The data presented in this study are available on request from the corresponding author. The data are not publicly available due to confidentiality.

**Acknowledgments:** We thank all the funders for their support and the valuable contributions of all investigators and participants.

**Conflicts of Interest:** The authors declare no conflict of interest.

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
