# Peer review of "Effect of Exergame Training on Working Memory and Executive Function in Older Adults"

_sustainability, doi:10.3390/su141710631_

Round 1

Reviewer 1 Report

Thank you for inviting me to review this manuscript on hot topic:

 Effect of exergame training on working memory and executive function in older adults. The manuscript is well written, and the results are very exciting and convincing.

I can propose several points:

1.     Please extend discussions, here I think you must compare your data with data from literature

2.     Please present more limitations of your study

Reviewer 2 Report

Overall: This paper could be more clearly written, having many awkward sentences and word choices that do not effectively convey the message.  I think it would benefit greatly from professional editing.

Abstract:

- Overall is well written and communicates the theme of the paper.  

- Please reconsider the third sentence beginning with "To date...". This sentence is awkward and disrupts the flow of the abstract.  I see what is meant by this, but it needs to be made clearer. 

Introduction:

- First sentence is very awkward.   I understand what you are trying to convey, but it should be stated more clearly.  

- Line 38: While it is true that it is important to prevent or reduce the risk of cognitive decline, your paper did not seek to do this.  There seems to be disconnect between this Introduction and the specifics of what was studied.  I do not gather that this study was conducted to see if exergaming would delay onset of dementia or reduce the burden among those with existing cognitive decline.  Rather, I perceive that it was conducted to see if exergaming can enhance specific domains of cognitive function among those without apparent cognitive impairment. 

- I think it is important to understand that the accepted convention for naming people over 65 years of age is "older adults".  "Elderly" is not mostly seen as negative or stereotypical.  Perhaps consider changing this.

- Line 43: You reference a stroke study to demonstrate that physical activity in important to cognition.  There are many other more directly relevant studies that should be referenced to reinforce a key element of your study.

- Line 45: How are you defining "long-term aerobic exercise"? 

- Line 47: "Even an acute...." I think you are missing the word "bout".

- Line 51: I think you need a different word from "approved".  Perhaps "shown" or "demonstrated".  The word "approved" makes me think that dual-task training of some kind has be approved as Tx for cognitive decline.

- My mentioning of dual-task training leads me to think you ought to talk about it or at least pull from existing literature.  Exergaming can be used to deliver cognitive-motor dual-task training.  There is a lot of very good and valuable literature in the dual-task exercise space that has not be highlighted in this paper.

Materials and Methods

- Was pre-exercise BP measured to ensure participants were not at elevated risk of an adverse event? I understand the PARQ was administered but this may not be sufficient given the age of participants.  Was any screening done before each exercise session to be sure safety status of participants did not change over time?

- If the exergaming session was group-based, how was activity tailored to ensure EACH participant was able to remain in their appropriate exercise intensity range? If a participant reached an exercise intensity above the pre-determined range, what was the action taken to address this?  

- The narrow selection of cognitive assessments is a limitation.

- Who administered the exercise sessions? What were their credentials, education, and training? 

- Were the participants given an orientation to the Switch games? How was the use of technology received? This is important information and could help others design impactful and manageable interventions that leverage gaming to enhance function in aging.  More details would be very helpful.

Results:

- Line 197: Generally I would argue against stating there is an effect at all, if the p-value is not significant.  

Discussion:

- This section as written is very lengthy and dives into content that is not that relevant to the current study.  While the fNIRS studies are interesting, I wonder if this content is a distraction.

- The narrow selection of cognitive domains assessed in this study is an important limitation.

- Your control group was not an attention control group.  This is a limitation, as time spent engaged could have a differential impact of participants' cognitive function between groups.

- I would love to see more methodological and study design details included, that will help others design similar studies.  This would add important content to the literature base.

- I would argue that not using brain imaging in this study is not necessarily a limitation.  Rather it seeks to answer a different question.

- Please seek professional editing for this paper.  There are many instances wherein word choices or sentence structure is confusing.

Reviewer 3 Report

Excellent work!

Please find attached my specific comments and suggestions.

Round 2

Reviewer 2 Report

I see that many changes have been made that do improve this paper.  However, there are some changes remaining that would improve the paper further.  

1) Please provide citations that physical activity levels have declined during COVID among older adults. 

2) While the addition of the dual-task literature is an improvement, the description is awkwardly written and disjointed.  This needs revision.  Lines 45 - 57 really need to be written more clearly.

3) There is still mention of "elderly" people.  I think this should be changed to "older adults".

4) Lines 254 to 263 are new.  Again, this section needs to be revised to be clearer.  It is very disjointed.

Round 3

Reviewer 2 Report

With regard to the references related to the decline in physical activity among older adults, during COVID-19, I see no references where they out to be.  Your first statement of the Introduction is an assumption without references.

Line 254: The sentence beginning with "The positive results..."I very awkwardly written.  

You state in the Results Section that small to moderate effects were observed for the ET group in both n-back conditions, relative to the ADT group, but most of your p-values are not significant.  Please explain this finding within the context of your interpretation. 
